# Glauert's optimum rotor disk revisited – a calculus of variations solution and exact integrals for thrust and bending moment coefficients

**Divya Tyagi and Sven Schmitz**

Department of Aerospace Engineering, The Pennsylvania State University, University Park, 16801, PA, USA

**Correspondence:** Sven Schmitz (sus52@psu.edu)

**Abstract.** The present work is an amendment to Glauert's optimum rotor disk solution for the maximum power coefficient, $C_{P_{max}}$, as a function of tip speed ratio, $\lambda$. First, an alternate mathematical approach is pursued towards the optimization problem by means of calculus of variations. Secondly, analytical solutions for thrust and bending moment coefficients, $C_T$ and $C_{Be}$, are derived, where an interesting characteristic is revealed pertaining to their asymptotic behavior for $\lambda \rightarrow \infty$. In addition, the limit case of the non-rotating actuator disk for $\lambda \rightarrow 0$ is shown for all three performance coefficients by repeated use of L'Hôpital's theorem, and its validity is discussed in the context of other works since Glauert.

## 1 Introduction

The original work of Glauert consisted of a closed-form solution describing the optimum performance of a rotating actuator disk (Glauert, 1935), where the simplified actuator disk concept for infinitely bladed propellers (Betz, 1919; Lanchester, 1915) was extended to wake rotation of energy-extracting rotors; see also Okulov and van Kuik (2012) for a review on the early history of momentum theory. In order for Glauert to close the basic equations of rotor disk theory, additional assumptions had to be applied that utilized a simple relationship between the induced velocity components and pressure jump at the rotor disk (Sørensen, 2016). At the time of Glauert, it was assumed that the kinetic energy required for wake rotation was extracted from the freestream (Goldstein, 1929), and this assumed wake energy and a constant pressure jump across the rotor disk remain the fundamental assumptions in Glauert's theory. In this regard, Glauert's model differed from that of Joukowski (1918), which was built on the assumption of constant circulation across the rotor disk (Burton et al., 2011; Sørensen, 2016). Also, de Vries (1979) questioned the validity of ignoring the static pressure drop caused by wake rotation. Indeed, an additional pressure drop is required to balance the centrifugal force of rotating air parcels

as a function of radial location on the disk; see also Sharpe (2004) for a discussion of a general momentum theory. A complete and comprehensive summary of the validity of all rotor disk theory models can be found in Sørensen (2016). In the end, various rotor disk models all approach the accepted Betz limit for maximum rotor power coefficient at high tip speed ratio. To date, however, no exact integrals have been presented for thrust and bending moment coefficients based on Glauert's optimum solution for axial and angular induction factors.

The objective of the present work is to determine the exact integrals for thrust and bending moment coefficients, serving as an addendum to Glauert's original work deriving optimum power coefficients. In doing so, an elegant solution by means of calculus of variation reveals itself, recovering Glauert's optimum distributions for axial and angular induction factors. Exact integrals for thrust and bending moment coefficients are then derived, including proper limit behavior of performance coefficients for low and high tip speed ratio. This work is organized as follows: Sect. 2 introduces classical rotor disk theory and relevant dimensionless parameters. Section 3 discusses Glauert's optimum solution for the maximum power coefficient, as well as an alternate optimization approach by means of calculus of variation, and a for-

mal asymptotic behavior of induction factors. Sections 4 and 5 detail the mathematics uncovering the exact integrals for thrust and bending moment coefficients, respectively. Section 6 summarizes the analytical solutions for all three performance coefficients and discusses the limitations with respect to validity at very low tip speed ratio. Section 7 contains some concluding remarks highlighting results obtained within the context of the typical range of tip speed ratio for modern wind turbines.

## 2   Rotor disk theory – axial/angular induction factors and power coefficient

The classical rotor disk formulation according to Glauert (1935) has been documented in various texts (Wilson et al., 1976; Hansen, 2008; Manwell et al., 2009; Burton et al., 2011; Wood, 2011; Schaffarczyk, 2014; Sørensen, 2016; Schmitz, 2019). In the following, only the primary relations relevant to this work are summarized.

Consider an axi-symmetric streamtube model that encompasses a wind turbine. The cross section, where the rotor is located, can be represented by a thin rotor disk of area, $A = \pi R^2$, where $R$ is the disk radius. The axial induction factor, $a$, determines the reduction in wind speed, $V_0$, at the disk. This coefficient is defined as

$$a = 1 - \frac{u}{V_0}, \tag{1}$$

where $u$ is the axial speed at the rotor disk. As a consequence of rotor thrust, there exists a discontinuity in pressure, $\Delta p$, at the disk. Therefore, Bernoulli's equation is applied both upstream and downstream of the disk. Adding these two equations together allows solving for $\Delta p$, i.e. the pressure jump arising from disk theory,

$$\Delta p = 2\rho V_0^2 a(1-a), \tag{2}$$

in terms of the fluid density, $\rho$, $V_0$, and $a$ as shown. Next, disk rotation is added at an angular speed, $\Omega$. The wake angular velocity component, $\omega$, just downstream of the disk, however, is affected by a rotor disk torque in the circumferential direction. The angular induction factor, $a'$, relates the wake angular velocity to the angular speed of the rotor disk:

$$a' = \frac{\omega}{2\Omega}. \tag{3}$$

Bernoulli's equation is applied once more upstream and downstream of the rotor disk. Note that in Glauert's theory, the assumption is that the same pressure jump, $\Delta p$, that generates thrust also generates rotor torque and power. The resulting $\Delta p$ equation arising from added wake rotation becomes

$$\Delta p = 2\rho V_0^2 a'(1+a')\lambda_r^2. \tag{4}$$

A practical relation between $a$ and $a'$ is obtained by equating Eqs. (2) and (4) such that

$$\lambda_r^2 = \frac{a(1-a)}{a'(1+a')}, \tag{5}$$

where $\lambda_r = \frac{r}{R}$ is the local tip speed ratio, with $\frac{r}{R}$ being the non-dimensional blade radius and $\lambda = \frac{\Omega R}{V_0}$ being the tip speed ratio. In rotor disk theory, the rotor power coefficient, $C_P = P/(\frac{1}{2}\rho V_0^2 A)$, with $P$ being rotor power, is obtained by the following integral:

$$C_P = \frac{8}{\lambda^2} \int_0^\lambda a'(1-a)\lambda_r^3 \, d\lambda_r. \tag{6}$$

## 3   Glauert's original optimum solution

In 1935, aerodynamicist Hermann Glauert approached the mathematical optimization problem of maximizing $C_P$ as a function of $\lambda$. Glauert's formal definition of the objective function $f$ is given as

$$f(a, a') = a'(1-a). \tag{7}$$

For the sake of brevity, Glauert's detailed derivation is not presented here but is included in modified form in Appendix A. It concludes in a third-order polynomial for $a(\lambda_r)$:

$$16a^3 - 24a^2 + (9 - 3\lambda_r^2)a + (\lambda_r^2 - 1) = 0, \tag{8}$$

which can be solved iteratively using, for example, a Newton–Raphson algorithm. Glauert also found a simple expression for $a'(a)$ that reads

$$a' = \frac{1 - 3a}{4a - 1}. \tag{9}$$

Next, Glauert substituted the optimum solutions for $a(\lambda_r)$ and $a'(a)$ back into Eq. (6) and solved for the exact integral for the maximum power coefficient, $C_{P_{max}}$, as a function of tip speed ratio, $\lambda$:

$$C_{P_{max}} = \frac{1}{\lambda^2} \cdot \left(\frac{2}{9}\right)^3$$
$$\left[\frac{64}{5}x^5 + 72x^4 + 124x^3 + 38x^2 - 63x - 12\ln x - \frac{4}{x}\right]_{x_2=1-3a_2}^{x_1=\frac{1}{4}}. \tag{10}$$

where $a_2$ is the corresponding solution of $a(\lambda_r)$ when evaluating Eq. (8) at $\lambda$. The exact $C_{P_{max}}$ integral from Eq. (10) converges to the theoretical Betz limit at $\frac{16}{27} \approx 0.5926$ for high $\lambda$. The reader is referred to Appendix A for details.

In the present paper, Glauert's work is amended first by finding a simple alternative solution based on calculus of variations and formally showing the behavior for low and high $\lambda$ and then by deriving corresponding exact integrals for thrust and bending moment coefficients, $C_T$ and $C_{Be}$.

## 3.1 A calculus of variations solution for $C_{P_{max}}$

A Lagrangian function, $\mathcal{L}(a, a', \mathcal{X})$, is defined as

$$\mathcal{L}(a, a', \mathcal{X}) = f(a, a') + \mathcal{X}g(a, a'), \tag{11}$$

where $f(a, a') = a'(1 - a)$ is the objective function from Eq. (7), $g(a, a') = a(1 - a) - a'(1 + a')\lambda_r^2$ is the equality constraint from Eq. (5), and $\mathcal{X}$ is the Lagrange multiplier. To maximize $f(a, a')$ under the equality constraint of $g(a, a') = 0$, the stationary points of $\mathcal{L}(a, a', \mathcal{X})$ must be determined by setting all partial derivatives of $\mathcal{L}$ with respect to $a$, $a'$, and $\mathcal{X}$ equal to 0. Those partial derivatives become

$$\frac{\partial \mathcal{L}}{\partial a} = -a' + \mathcal{X}(1 - 2a) = 0, \tag{12}$$

$$\frac{\partial \mathcal{L}}{\partial a'} = 1 - a - \mathcal{X}\lambda_r^2(1 + 2a') = 0, \tag{13}$$

$$\frac{\partial \mathcal{L}}{\partial \mathcal{X}} = a(1 - a) - a'(1 + a')\lambda_r^2 = 0. \tag{14}$$

The goal is to solve this system of equations for polynomials $a(\lambda_r)$ and $a'(\lambda_r)$. This can be done multiple ways; however, a simple method involves equating $\mathcal{X}$ from Eqs. (12) and (13). The resulting expression is then substituted into the final partial derivative in Eq. (14). After some algebraic manipulation, a factored polynomial for $a(\lambda_r)$ reads

$$(16a^3 - 24a^2 + (9 - 3\lambda_r^2)a + (\lambda_r^2 - 1)) \cdot (a - 1) = 0, \tag{15}$$

where its factor of third order recovers Glauert's original polynomial from Eq. (8).

The same system of equations can be solved instead to compute a polynomial for $a'(\lambda_r)$. This time, $a'$ from Eqs. (12) and (13) is equated, and the resulting expression for $a$ is then substituted into the partial derivative from Eq. (14). After some algebraic simplification, a third-order polynomial for $a'(\lambda_r)$ is obtained:

$$16\lambda_r^2(a')^3 + 24\lambda_r^2(a')^2 + (9\lambda_r^2 - 3)a' - 2 = 0. \tag{16}$$

The relations presented in Eqs. (15) and (16) were computed by means of calculus of variations, a methodology different from Glauert's original approach. However, both approaches produce identical results for the optimum flow conditions. Note that a calculus of variations approach for lightly loaded propellers has been documented in Breslin and Andersen (1994).

## 3.2 Limiting case of $a$ & $a'$ for low and high tip speed ratio

Next, it is of interest to determine the limiting case for $a$ as $\lambda_r$ tends to both 0 and $\infty$. As $\lambda_r \to 0$, Eq. (15) becomes

$$16a^3 - 24a^2 + 9a - 1 = (a - 1)(4a - 1)^2 = 0, \tag{17}$$

which has the trivial roots $a = \frac{1}{4}$, 1. Here, the physical solution of $a = \frac{1}{4}$ is consistent with Glauert's original work. As for the upper limit of $\lambda_r \to \infty$ for Eq. (15), the equation can be recast to obtain

$$\frac{1}{1 + \lambda_r^2} = \frac{1 - 3a}{-2(2a - 1)^3}, \tag{18}$$

where it becomes apparent that as the left-hand side of Eq. (18) tends to zero for $\lambda_r \to \infty$, the right-hand side can only reconcile this for $a \to \frac{1}{3}$. This result is again consistent with Glauert's original solution. Alternatively, the factorization in Eq. (17) can be used to restate the third-order factor in Eq. (15) to obtain

$$\frac{1}{\lambda_r^2} = \frac{1 - 3a}{(1 - a)(1 - 4a)^2}, \tag{19}$$

which concludes the same for $\lambda_r \to 0, \infty$ and is in fact a relation used by Glauert in the exact integral for $C_{P_{max}}$ (see Appendix A) and is used in Sect. 3 for the exact integrals of $C_T$ and $C_{Be}$. For completeness, the limiting case for $a'$ as $\lambda_r$ tends to both 0 and $\infty$ is also determined. Equation (16) is rearranged to

$$\frac{1}{\lambda_r^2} = \frac{(4a' + 3)^2}{2 + 3a'}a'. \tag{20}$$

For added wake rotation $a' > 0$, the left-hand side of Eq. (20) tends to $\infty$ for $\lambda_r \to 0$. This can be reconciled on the right-hand side by $a' \to \infty$. Likewise, as the limit of the left-hand side of Eq. (20) becomes 0 for $\lambda_r \to \infty$, the right-hand side will only be satisfied with a physical solution of $a' \to 0$. Both behaviors are consistent with Glauert's solution (see Appendix A).

## 3.3 Limiting case of $C_{P_{max}}$ for low and high tip speed ratio

The limiting case of the $C_{P_{max}}$ integral for $\lambda_r \to 0$ is not easily shown and in fact was not explicitly stated in Glauert's original work. It is added here as part of the amendment. To better illustrate the behavior, the substitution $x = 1 - 3a$ is used along with the practical relation from Eq. (19) at the integration bound $a_2$ such that Eq. (10) becomes

$$C_{P_{max}} = \frac{(1 - 3a_2)}{(1 - a_2)(1 - 4a_2)^2} \cdot \left(\frac{2}{9}\right)^3$$

$$\left[-10.5082 - \left(\frac{64}{5}(1 - 3a_2)^5 + 72(1 - 3a_2)^4 + 124(1 - 3a_2)^3\right.\right.$$

$$\left.\left. + 38(1 - 3a_2)^2 - 63(1 - 3a_2) - 12\ln(1 - 3a_2) - \frac{4}{(1 - 3a_2)}\right)\right]. \tag{21}$$

It is evident that for $\lambda_r = 0$, where $a_2 = \frac{1}{4}$, there exists a singularity for $C_{P_{max}}$. Therefore, a limit for $C_{P_{max}}$ as $\lambda_r \to 0$, or as $a_2 \to \frac{1}{4}$, must be taken and results in the following:

$$\lim_{a_2 \to \frac{1}{4}} C_{P_{max}} = \frac{0}{0}. \tag{22}$$

Since evaluating this limit results in the indeterminate form of $\frac{0}{0}$, the mathematical theorem known as L'Hôpital's rule can be applied to determine the true limit using derivatives:

$$\lim_{a_2 \to \frac{1}{4}} \frac{f(a_2)}{g(a_2)} = \lim_{a_2 \to \frac{1}{4}} \frac{f'(a_2)}{g'(a_2)} = \lim_{a_2 \to \frac{1}{4}} \frac{f''(a_2)}{g''(a_2)}. \tag{23}$$

For ease of reference, the functions $f$ and $g$ extracted from Eq. (21) are explicitly stated below:

$$f(a_2) = \left(\frac{2}{9}\right)^3 \Big[ -10.5082 - \frac{64}{5}(1-3a_2)^5 + 72(1-3a_2)^4$$
$$+ 124(1-3a_2)^3 + 38(1-3a_2)^2 - 63(1-3a_2)$$
$$- 12\ln(1-3a_2) - \frac{4}{(1-3a_2)} \Big], \tag{24}$$

$$g(a_2) = \frac{(1-a_2)(1-4a_2)^2}{1-3a_2}. \tag{25}$$

Applying the limit of $a_2 \to \frac{1}{4}$ (or $\lambda_r \to 0$) does result in the following:

$$\lim_{a_2 \to \frac{1}{4}} \frac{f'(a_2)}{g'(a_2)}$$

$$= \lim_{a_2 \to \frac{1}{4}} \frac{\frac{24(64a_2^6 - 224a_2^5 + 308a_2^4 - 212a_2^3 + 77a_2^2 - 14a_2 + 1)}{(3a_2-1)^2}}{\frac{6(4a_2-1)(4a_2^2 - 4a_2 + 1)}{(3a_2-1)^2}} = \frac{0}{0}. \tag{26}$$

Applying L'Hôpital's rule twice, however, proves the following:

$$\lim_{a_2 \to \frac{1}{4}} \frac{f''(a_2)}{g''(a_2)}$$

$$= \lim_{a_2 \to \frac{1}{4}} \frac{96(192a_2^6 - 600a_2^5 + 742a_2^4 - 467a_2^3 + 159a_2^2 - 28a_2 + 2)}{12(24a_2^3 - 24a_2^2 + 8a_2 - 1)}$$

$$= 0. \tag{27}$$

Thereby the intuitive result that $C_{P_{\max}} \to 0$ as $\lambda_r \to 0$ has been formally shown. Note that the high tip speed ratio limit of Eq. (21) for $a_2 \to \frac{1}{3}$ (or $\lambda_r \to \infty$) is more readily shown with

$$\lim_{a_2 \to \frac{1}{3}} C_{P_{\max}} = \lim_{a_2 \to \frac{1}{3}} \frac{4}{(1-a_2)(1-4a_2)^2} \cdot \left(\frac{2}{9}\right)^3 = \frac{16}{27}, \tag{28}$$

which indeed recovers the known Betz limit. Next follows an additional amendment to Glauert's work by means of analytical derivations for thrust and bending moment coefficients, $C_T$ and $C_{Be}$, based on optimum $a$ and $a'$ distributions.

## 4 Exact integral of the thrust coefficient $C_T$ based on Glauert's optimum solution

The pressure jump, $\Delta p$, generates a rotor thrust, $T = \Delta p\, A$, across the rotor disk in the axial direction. In differential form, this becomes $dT = \Delta p\, dA = 4\pi\rho V_0^2 a(1-a)r\,dr$, using the definition of $\Delta p$ from Eq. (2) and $dA = 2\pi r\,dr$. A dimensionless rotor thrust coefficient is then defined as

$$C_T = \frac{T}{\frac{1}{2}\rho V_0^2 A}. \tag{29}$$

The incremental thrust coefficient, $dC_T$, can be computed by dividing the incremental thrust, $dT$, by $\frac{1}{2}\rho V_0^2 A$, resulting in the following expression for $dC_T$:

$$dC_T = 8a(1-a)\frac{r}{R}d(\frac{r}{R}). \tag{30}$$

Recall that the definition for local tip speed ratio is $\lambda_r = \frac{r}{R}\lambda$. Substituting the expression for non-dimensional blade radius, $\frac{r}{R}$, into Eq. (30) leads to the final expression for $dC_T$:

$$dC_T = \frac{8}{\lambda^2}a(1-a)\lambda_r d\lambda_r. \tag{31}$$

To write $C_T$ exclusively in terms of $a$, differentiating Eq. (19) yields a relation for $\lambda_r d\lambda_r$ with

$$\lambda_r d\lambda_r = \frac{3(4a-1)(1-2a)^2}{(1-3a)^2}da, \tag{32}$$

such that an integral for $C_T$ can be written as

$$C_T = \frac{8}{\lambda^2} \int_0^\lambda a(1-a)\lambda_r\, d\lambda_r$$

$$= -\frac{24}{\lambda^2} \int_0^\lambda \frac{a(1-a)(1-4a)(1-2a)^2}{(1-3a)^2}da. \tag{33}$$

The same substitution as used by Glauert is carried through, where $x = 1 - 3a$, resulting in

$$C_T = \frac{1}{\lambda^2} \cdot \frac{8}{243} \int_{x_2}^{x_1} \Big[ \frac{(1-x)(2+x)(1-4x)(1+2x)^2}{x^2} \Big] dx$$

$$= -\frac{1}{\lambda^2} \cdot \frac{8}{243} \int_{x_1}^{x_2} \Big( 16x^3 + 28x^2 - 20x - 25 - \frac{1}{x} + \frac{2}{x^2} \Big) dx, \tag{34}$$

which can be easily integrated to yield the following:

$$C_T = \frac{1}{\lambda^2}$$

$$\cdot \frac{8}{243} \Big[ 4x^4 + \frac{28}{3}x^3 - 10x^2 - 25x - \ln x - \frac{2}{x} \Big]_{x_2 = 1 - 3a_2}^{x_1 = \frac{1}{4}}.$$

(35)

To better understand the behavior of the $C_T$ integral, Eq. (35) is rewritten again in terms of $a$. The integration bounds $x_1$ and $x_2$ are substituted in as $\frac{1}{4}$ and $(1 - 3a_2)$, respectively, where $\lambda^2 = \lambda_r^2|_{a_2}$ such that

$$C_T = \frac{(1 - 3a_2)}{(1 - a_2)(1 - 4a_2)^2} \cdot \frac{8}{243} \Bigg[ -13.3272 - \Big(4(1 - 3a_2)^4$$

$$+ \frac{28}{3}(1 - 3a_2)^3 - 10(1 - 3a_2)^2 - 25(1 - 3a_2)$$

$$- \ln(1 - 3a_2) - \frac{2}{(1 - 3a_2)}\Big)\Bigg].$$

(36)

As $\lambda_r \to 0$, or $a_2 \to \frac{1}{4}$, there exists a singularity and $C_T$ is not defined (note: a similar behavior was found earlier for $C_{P_{max}}$). Indeed the limit for $C_T$ as $\lambda_r \to 0$, or $a_2 \to \frac{1}{4}$, becomes

$$\lim_{a_2 \to \frac{1}{4}} C_T = \frac{0}{0}.$$

(37)

Since evaluating this limit results in the indeterminate form of $\frac{0}{0}$, L'Hôpital's rule can be applied to determine the true limit. The theorem equates the following limits, where functions $f$ and $g$ are differentiable:

$$\lim_{a_2 \to c} \frac{f(a_2)}{g(a_2)} = \lim_{a_2 \to c} \frac{f'(a_2)}{g'(a_2)}.$$

(38)

For ease of reference, the functions $f$ and $g$ extracted from Eq. (36) are explicitly stated below:

$$f(a_2) = \frac{8}{243} \Bigg[ -13.3272 - \Big(4(1 - 3a_2)^4 + \frac{28}{3}(1 - 3a_2)^3$$

$$- 10(1 - 3a_2)^2 - 25(1 - 3a_2) - \ln(1 - 3a_2)$$

$$- \frac{2}{(1 - 3a_2)}\Big)\Bigg],$$

(39)

$$g(a_2) = \frac{(1 - a_2)(1 - 4a_2)^2}{1 - 3a_2}.$$

(40)

Applying L'Hôpital's rule once results in the following as $a_2$ approaches $\frac{1}{4}$:

$$\lim_{a_2 \to \frac{1}{4}} \frac{f'(a_2)}{g'(a_2)} = \lim_{a_2 \to \frac{1}{4}} \frac{\frac{-24a_2(16a_2^4 - 36a_2^3 + 28a_2^2 - 9a_2 + 1)}{(3a_2 - 1)^2}}{\frac{6(4a_2 - 1)(4a_2^2 - 4a_2 + 1)}{(3a_2 - 1)^2}}$$

$$= \frac{0}{0}.$$

(41)

It is valid to apply L'Hôpital's rule a second time, as functions $f$ and $g$ are differentiable. This yields

$$\lim_{a_2 \to \frac{1}{4}} \frac{f''(a_2)}{g''(a_2)} = \lim_{a_2 \to \frac{1}{4}} \frac{-80a_2^4 + 144a_2^3 - 84a_2^2 + 18a_2 - 1}{12a_2^2 - 10a_2 + 2}$$

$$= \frac{3}{4},$$

(42)

where it is interesting to note that the thrust coefficient, $C_T$, converges to 0.75 as $\lambda \to 0$. Though seemingly a surprising result at first, it will be reconciled with actuator disk theory in a later section.

Note again that the high tip speed ratio limit of Eq. (36) for $\lambda_r \to \infty$ (or $a_2 \to \frac{1}{3}$) is more readily shown with

$$\lim_{a_2 \to \frac{1}{3}} C_T = \lim_{a_2 \to \frac{1}{3}} \frac{2}{(1 - a_2)(1 - 4a_2)^2} \cdot \frac{8}{243} = \frac{8}{9},$$

(43)

which is also fully consistent with actuator disk theory, as will be shown further below.

## 5 Exact integral of the bending moment coefficient $C_{Be}$ based on Glauert's optimum solution

The bending moment, Be, is an important structural parameter when assessing the loading of wind turbine blades. A dimensionless bending moment coefficient is defined as

$$C_{Be} = \frac{Be}{\frac{1}{2}\rho V_0^2 AR}.$$

(44)

In differential form, dBe is essentially the product of thrust $dC_T$ and lever arm $\frac{r}{R}$ of the local annulus such that

$$dC_{Be} = dC_T \cdot \frac{r}{R}$$

$$= \frac{8}{\lambda^2} a(1 - a)\lambda_r d\lambda_r \cdot \frac{r}{R}$$

$$= \frac{8}{\lambda^3} a(1 - a)\lambda_r^2 d\lambda_r,$$

(45)

where Eq. (31) was used. The exact integral for the total bending moment coefficient, $C_{Be}$, is hence defined as

$$C_{Be} = \int_0^\lambda dC_{Be} = \frac{8}{\lambda^3} \int_0^\lambda a(1 - a)\lambda_r^2 d\lambda_r.$$

(46)

Substituting a combination of Eqs. (19) and (32) yields

$$C_{Be} = -\frac{24}{\lambda^3} \int_0^\lambda \frac{a(1 - a)^{3/2}(1 - 4a)^2(1 - 2a)^2}{(1 - 3a)^{5/2}} da.$$

(47)

Once more, the integration substitution is performed, where $x = 1 - 3a$, so that $C_{Be}$ can be solved analytically to

$$C_{Be} = \frac{1}{\lambda^3} \cdot \frac{8}{243 \cdot 27^{1/2}} \int_{x_1}^{x_2} \frac{(1-x)(2+x)^{3/2}}{(1-4x)^2(1+2x)^2} \frac{1}{x^{5/2}} dx$$

$$= \frac{1}{\lambda^3} \cdot \frac{8}{243 \cdot 27^{1/2}} \left[ -24\ln((x+2)^{1/2} + x^{1/2}) \right.$$

$$\left. - \frac{(x+2)^{1/2}(192x^6 + 408x^5 - 532x^4 - 890x^3 + 585x^2 - 260x + 20)}{15x^{3/2}} \right]_{x_2=1-3a_2}^{x_1=\frac{1}{4}}. \quad (48)$$

To better understand the behavior of the $C_{Be}$ function, Eq. (48) is rewritten solely in terms of $a$. This means substituting $\lambda^3 = \lambda_r^3|_{a_2}$ in the denominator by raising Eq. (19), which is evaluated at the integration bound $a_2$, to the power $\frac{3}{2}$ such that

$$\lambda^3 = -\frac{(1-a_2)^{\frac{3}{2}}(1-4a_2)^3}{(1-3a_2)^{\frac{3}{2}}}. \quad (49)$$

The values for $x_1$ and $x_2$ are substituted in as well, being $\frac{1}{4}$ and $(1-3a_2)$, respectively, where $a_2$ is simply $a(\lambda_r)$ such that

$$C_{Be} = -\frac{(1-3a_2)^{3/2}}{(1-a_2)^{3/2}(1-4a_2)^3} \cdot \frac{8}{243 \cdot 27^{1/2}}$$

$$\left[ 2.5457 - \left( -24\ln[(3-3a_2)^{1/2} + (1-3a_2)^{1/2}] \right. \right.$$

$$- \frac{(3-3a_2)^{1/2} \cdot (192(1-3a_2)^6 + 408(1-3a_2)^5 - 532(1-3a_2)^4 - 890(1-3a_2)^3)}{15 \cdot (1-3a_2)^{3/2}}$$

$$\left. \left. - \frac{(3-3a_2)^{1/2} \cdot (585(1-3a_2)^2 - 260(1-3a_2) + 20)}{15 \cdot (1-3a_2)^{3/2}} \right) \right]. \quad (50)$$

As $\lambda_r \to 0$ (or $a_2 \to \frac{1}{4}$), the bending moment coefficient, $C_{Be}$, yields the following indeterminate form:

$$\lim_{a \to \frac{1}{4}} C_{Be} = \frac{f(a_2)}{g(a_2)} = \frac{0}{0}. \quad (51)$$

For ease of reference, the functions $f$ and $g$ extracted from Eq. (50) are explicitly stated below:

$$f(a) = \frac{8}{243 \cdot 27^{1/2}}$$

$$\left[ 2.5457 - \left( (-24\ln[(3-3a_2)^{1/2} + (1-3a_2)^{1/2}] \right. \right.$$

$$- \frac{(3-3a_2)^{1/2} \cdot (192(1-3a_2)^6 + 408(1-3a_2)^5 - 532(1-3a_2)^4 - 890(1-3a_2)^3)}{15 \cdot (1-3a_2)^{3/2}}$$

$$\left. \left. - \frac{(3-3a_2)^{1/2} \cdot (585(1-3a_2)^2 - 260(1-3a_2) + 20)}{15 \cdot (1-3a_2)^{3/2}} \right) \right], \quad (52)$$

$$g(a) = -\frac{(1-a_2)^{3/2}(1-4a_2)^3}{(1-3a_2)^{3/2}}. \quad (53)$$

Using L'Hôpital's rule once leads to the indeterminate form of $\frac{0}{0}$ with

$$f'(a_2) = \frac{\begin{array}{c} 3359232a_2^7 - 11757312a_2^6 + 16166304a_2^5 - 11127456a_2^4 \\ + 4041576a_2^3 - 734832a_2^2 + 52488a_2 \end{array}}{3^{\frac{13}{2}}(1-3a_2)^{\frac{5}{2}}(3-3a_2)^{1/2}},$$

$$g'(a_2) = \frac{9(1-a_2)^{1/2}(4a_2-1)^2(4a_2^2-4a_2+1)}{(1-3a_2)^{\frac{5}{2}}},$$

and

$$C_{Be} = \lim_{a_2 \to \frac{1}{4}} \frac{f'(a_2)}{g'(a_2)} = \frac{0}{0}. \quad (54)$$

Applying L'Hôpital's rule a second time still results in the indeterminate form of $\frac{0}{0}$ as

$$f''(a_2) = \frac{\begin{array}{c} 120932352a_2^8 - 519001344a_2^7 + 925888320a_2^6 - 893765664a_2^5 \\ + 509553504a_2^4 - 175414896a_2^3 + 35429400a_2^2 - 3779136a_2 + 157464 \end{array}}{3^{\frac{13}{2}}(1-3a_2)^{\frac{7}{2}}(3-3a_2)^{3/2}},$$

$$g''(a_2) = \frac{3456a_2^5 - 7776a_2^4 + 6624a_2^3 - 2736a_2^2 + 558a_2 - 45}{(1-3a_2)^{7/2}(1-a_2)^{1/2}},$$

and

$$C_{Be} = \lim_{a_2 \to \frac{1}{4}} \frac{f''(a_2)}{g''(a_2)} = \frac{0}{0}. \quad (55)$$

Differentiating functions $f$ and $g$ a third time results in a definite value for the limiting case of $C_{Be}$ with

$$f'''(a_2) = \frac{\begin{array}{c} 3265173504a_2^9 - 16597965312a_2^8 + 36147435840a_2^7 - 44231007744a_2^6 + 33530384160a_2^5 \\ - 16363658880a_2^4 + 5158048248a_2^3 - 1017532368a_2^2 + 114791256a_2 - 5668704 \end{array}}{3^{\frac{13}{2}}(1-3a_2)^{\frac{9}{2}}(3-3a_2)^{\frac{5}{2}}}$$

$$(56)$$

$$g'''(a_2) = \frac{\begin{array}{c} 10368a_2^6 - 31104a_2^5 + 36288a_2^4 \\ - 21312a_2^3 + 6642a_2^2 - 1026a_2 + 63 \end{array}}{(1-3a_2)^{\frac{9}{2}}(1-a_2)^{\frac{3}{2}}}, \quad (57)$$

and

$$C_{Be} = \lim_{a_2 \to \frac{1}{4}} \frac{f'''(a_2)}{g'''(a_2)} = \frac{1}{2}. \quad (58)$$

Note again that the high tip speed ratio limit of Eq. (50) for $\lambda_r \to \infty$ is more readily shown with

$$\lim_{a_2 \to \frac{1}{3}} C_{Be} = \lim_{a_2 \to \frac{1}{3}} \frac{-(3-3a_2)^{\frac{1}{2}} \cdot \frac{4}{3}}{(1-a_2)^{\frac{3}{2}}(1-4a_2)^3} \cdot \frac{8}{243 \cdot 27^{\frac{1}{2}}} = \frac{16}{27}. \quad (59)$$

For the high $\lambda_r$ limit, the value of $C_{Be}$ approaches $\frac{16}{27} \approx 0.5926$. Here it is interesting to note that both $C_{P_{max}}$ and $C_{Be}$ tend to the Betz limit as $\lambda_r \to \infty$, which is reconciled in the next section.

| $\lambda$ | $C_{P_{max}}$ | $C_T$ | $C_{Be}$ |
|---|---|---|---|
| $\to 0$ | 0 | 0.7500 | 0.5000 |
| 1 | 0.4155 | 0.8458 | 0.5685 |
| 2 | 0.5112 | 0.8689 | 0.5828 |
| 3 | 0.5454 | 0.8773 | 0.5874 |
| 4 | 0.5615 | 0.8812 | 0.5894 |
| 5 | 0.5704 | 0.8834 | 0.5905 |
| 6 | 0.5759 | 0.8847 | 0.5911 |
| 7 | 0.5795 | 0.8856 | 0.5914 |
| 8 | 0.5820 | 0.8863 | 0.5917 |
| 9 | 0.5838 | 0.8867 | 0.5919 |
| 10 | 0.5852 | 0.8871 | 0.5920 |
| $\to \infty$ | 0.5926 | 0.8889 | 0.5926 |

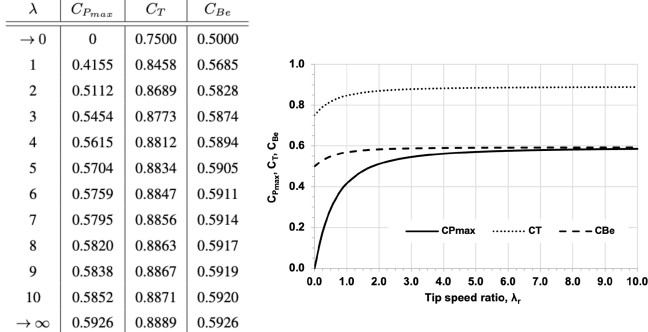

**Figure 1.** Power $C_{P_{max}}$, thrust $C_T$, and bending moment $C_{Be}$ coefficients as functions of tip speed ratio, $\lambda$, for optimal $a$ and $a'$ distributions.

## 6 Summary of coefficients derived from Glauert's optimum model

Figure 1 shows the three coefficients of interest, $C_{P_{max}}$, $C_T$, and $C_{Be}$, and their tabulated values over a range of design tip speed ratios between 0 and 10. Note that Glauert's original work showed exclusively $C_{P_{max}}$ based on optimum flow conditions. Exact analytical solutions for $C_T$ and $C_{Be}$, on the other hand, constitute an amendment to Glauert's original work.

Beginning with the high tip speed ratio limit of $\lambda \to \infty$, both power and bending moment coefficients, $C_{P_{max}}$ and $C_{Be}$, converge to the Betz limit $\frac{16}{27} \approx 0.5926$. This may at first seem surprising for $C_{Be}$; however, it is a direct consequence of the respective limit for the thrust coefficient, $C_T$. In fact, $C_T \to \frac{8}{9}$ for $\lambda \to \infty$, which describes the limit of actuator disk theory with $C_T = 4a(1-a) = \frac{8}{9}$ for the optimum $a \to \frac{1}{3}$. In addition, one consequence of a constant pressure jump, $\Delta p$, across the rotor disk is a linear thrust distribution $\mathrm{d}T = \Delta p\, 2\pi r\, \mathrm{d}r$, whose center of pressure is known to be at $\frac{2}{3}\frac{r}{R}$. From this simple thought experiment, it is indeed understood that $C_{Be} = \frac{2}{3}\, C_T = \frac{16}{27}$ (Betz limit) for $\lambda \to \infty$.

For the low tip speed ratio limit of $\lambda \to 0$, it is known that $C_{P_{max}} \to 0$; however, it has been formally proven for the first time as part of this amendment using L'Hôpital's theorem. On the other hand, thrust and bending moment coefficients, $C_T$ and $C_{Be}$, tend towards non-zero values. The thrust coefficient, $C_T$, remains consistent with a non-rotating actuator disk such that $C_T = 4a(1-a) = \frac{3}{4}$ for $a \to \frac{1}{4}$. The center of pressure stays at $\frac{2}{3}\frac{r}{R}$ in the limit such that $C_{Be} = \frac{2}{3}\, C_T = \frac{1}{2}$, all of which is consistent with Fig. 1 and the limit of actuator disk theory. Note, however, that for rotor disk theory where $0 < \lambda < \infty$, the ratio $C_{Be}/C_T \approx \frac{2}{3}$, though not exactly, as $a(\lambda_r) \neq$ constant in Glauert's solution (see Fig. A1).

### 6.1 Validity of classical Glauert theory

In his approach, Glauert assumed the pressure in the wake to be approximately equal to the freestream pressure and the azimuthal velocity in the wake to be approximately equal to the azimuthal velocity immediately behind the rotor plane (Sørensen, 2016). The first assumption has been questioned by others (de Vries, 1979) due to the pressure gradient required to maintain swirl. Generally, areas of improvement have been identified within Glauert's work for low tip speed ratio that include refinement of the blade model, swirl effects, tip correction, the optimization routine, and considerations for atmospheric and rotor conditions (van Kuik, 2018). In this context, the greatest variation among different aerodynamic rotor models is present within the optimum flow conditions, $a(\lambda_r)$ and $a'(\lambda_r)$, for low tip speed ratio ($\lambda < 4$). For example, Wood and Hammam (2022) investigated the optimal performance of actuator disk models for horizontal-axis turbines. By implementing thrust from the Kutta–Joukowski equation, dependence on pressure within the wake is avoided. Their results highlighted that optimal performance at low $\lambda$ is constrained as a measure to avoid recirculation in the wake – something that was not captured in Glauert's theory. While apparent effects on $C_{P_{max}}$ are small at low tip speed ratio, the thrust coefficient $C_T$ for the limit $\lambda \to 0$ computed by Wood and Hammam (2022) is less than half compared to the computed value of 0.75 in Fig. 1. At $\lambda = 1$, however, $C_T$ from Wood and Hammam (2022) equals 0.738 compared to 0.8458 in Fig. 1, and $C_{P_{max}}$ is 0.4381 compared to 0.4155 in Fig. 1. Overall, results obtained using Glauert's theory are impressive given the inherent assumptions.

## 7 Concluding remark

This work derived several amendments to Glauert's original optimum rotor disk solution. First, an alternative approach by means of calculus of variations was pursued to solve the underlying classical objective function for $C_{P_{max}}$ in wind turbine aerodynamics. Second, Glauert's optimum rotor disk solution was used to derive exact solutions for the thrust and bending moment coefficients, $C_T$ and $C_{Be}$. Here, L'Hôpital's theorem was employed to determine the convergence behavior of all three coefficients for $\lambda \to 0$, while the high tip speed ratio limit of $\lambda \to \infty$ was more readily shown. Some surprising results included that both power and bending moment coefficients, $C_{P_{max}}$ and $C_{Be}$, approach the Betz limit for $\lambda \to \infty$ and that thrust and bending moment coefficients, $C_T$ and $C_{Be}$, have non-zero values for $\lambda \to 0$. Using a simple thought experiment, it was shown that all observations are indeed consistent with the limit of a uniformly loaded (constant pressure jump) rotor disk, whose validity holds for tip speed ratios typical of modern utility-scale wind turbines but has, due to its inherent assumptions, limited validity at very low tip speed ratio as discussed. Nevertheless, this work presents

an interesting addendum to one of the pioneering works in wind turbine aerodynamics.

## Appendix A: Glauert's original $C_{P_{max}}$ derivation

The following analysis is adjusted from Glauert's original derivation (Glauert, 1935) and consistent with other versions published in various textbooks (Burton et al., 2011; Hansen, 2008; Manwell et al., 2009; Schaffarczyk, 2014; Sørensen, 2016; Wilson et al., 1976; Wood, 2011; Schmitz, 2019). The function to be optimized is

$$f(a, a') = a'(1 - a). \tag{A1}$$

To determine the maximum of $f$, one must differentiate both sides of the objective function $f$ with respect to the axial induction factor, $a$. The result is equated to 0 in order to find the appropriate stationary point, as shown:

$$\frac{df}{da} = \frac{da'}{da}(1 - a) - a' = 0. \tag{A2}$$

Rearranging the equation above yields a new condition which must be satisfied at maximum $C_P$:

$$\frac{da'}{da} = \frac{a'}{1 - a}. \tag{A3}$$

Now, a second look is taken at the pressure relation from Eq. (5) which was known to Glauert. A derivative with respect to $a$ is taken on the left- and right-hand sides of the equation, resulting in the following:

$$1 - 2a = \lambda_r^2(1 + 2a')\frac{da'}{da}. \tag{A4}$$

The right-hand side of Eq. (5) is substituted in for the $\lambda_r^2$ term, and the right-hand side of Eq. (A3) is substituted in for the differential term above, simplifying to the following:

$$\frac{1 + a'}{1 + 2a'} = \frac{a}{1 - 2a}. \tag{A5}$$

Upon algebraic rearranging of this intermediate step in Eq. (A5), the optimum relationship between $a$ and $a'$ is revealed:

$$a' = \frac{1 - 3a}{4a - 1}. \tag{A6}$$

This solution for $a'$ can be substituted into the pressure relation from Eq. (5) to obtain a relationship between $\lambda_r$ and $a$:

$$\lambda_r^2 = \frac{(1 - a)(1 - 4a)^2}{1 - 3a}. \tag{A7}$$

Equation (A7) is then rearranged into a third-degree polynomial representing the optimal axial induction factors

across the rotor disk as a function of local tip speed ratio, $\lambda_r$, as shown below:

$$16a^3 - 24a^2 + (9 - 3\lambda_r^2)a + (\lambda_r^2 - 1) = 0. \tag{A8}$$

A Newton–Raphson algorithm is employed to iteratively solve Glauert's third-order polynomial, with

$$a_{i+1} = a_i - \frac{f(a_i)}{f'(a_i)}. \tag{A9}$$

The resulting optimum function $a(\lambda_r)$ has been tabulated and plotted in Fig. A1 for $\lambda_r$ ranging from 0 to 10. A formal proof of $C_P$ exhibiting a maximum, i.e. via $\frac{d^2 f}{da^2} < 0$, has only been shown recently (Schmitz, 2019).

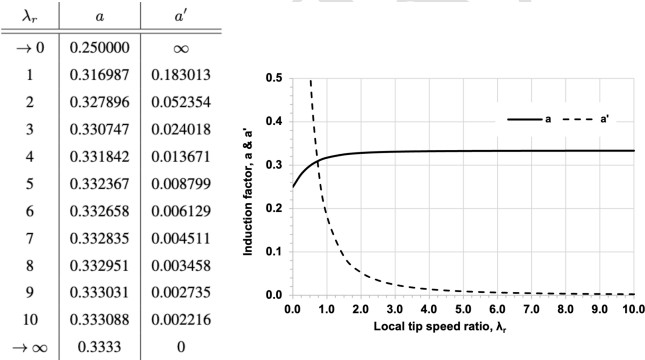

| $\lambda_r$ | $a$ | $a'$ |
|---|---|---|
| $\to 0$ | 0.250000 | $\infty$ |
| 1 | 0.316987 | 0.183013 |
| 2 | 0.327896 | 0.052354 |
| 3 | 0.330747 | 0.024018 |
| 4 | 0.331842 | 0.013671 |
| 5 | 0.332367 | 0.008799 |
| 6 | 0.332658 | 0.006129 |
| 7 | 0.332835 | 0.004511 |
| 8 | 0.332951 | 0.003458 |
| 9 | 0.333031 | 0.002735 |
| 10 | 0.333088 | 0.002216 |
| $\to \infty$ | 0.3333 | 0 |

**Figure A1.** Glauert's theoretical solutions for optimum axial and angular induction factors, $a$ and $a'$, respectively, as a function of local tip speed ratio, $\lambda_r$.

With optimum flow conditions known for $a$ and $a'$ as a function of $\lambda_r$, Glauert was able to determine the exact solution for $C_{P_{max}}$. Returning to Eq. (6), there is a $\lambda_r^3 d\lambda_r$ term that must be addressed in order to fully solve $C_{P_{max}}$ in terms of $a$. The approach taken by Glauert involved taking a second look at Eq. (A7). Upon differentiating both sides of this equation, a new expression for $2\lambda_r d\lambda_r$ is found, relating $d\lambda_r$ and $da$ such that

$$2\lambda_r d\lambda_r = \frac{6(4a - 1)(1 - 2a)^2}{(1 - 3a)^2} da. \tag{A10}$$

The $\lambda_r^3 d\lambda_r$ term of interest can then be broken into a $\lambda_r^2$ and $\lambda_r d\lambda_r$ term, for which Eqs. (A7) and (A10) can be substituted in. Now, the integral for the maximum power coefficient, $C_{P_{max}}$, can be defined by just one unknown, i.e. $a$, as shown below:

$$\begin{aligned} C_{P_{max}} &= \frac{8}{\lambda^2} \int_0^\lambda a'(1 - a)\lambda_r^2 \cdot \lambda_r \, d\lambda_r \\ &= \frac{24}{\lambda^2} \int_{a_1}^{a_2} \frac{(1 - a)^2(1 - 4a)^2(1 - 2a)^2}{(1 - 3a)^2} \, da. \end{aligned} \tag{A11}$$

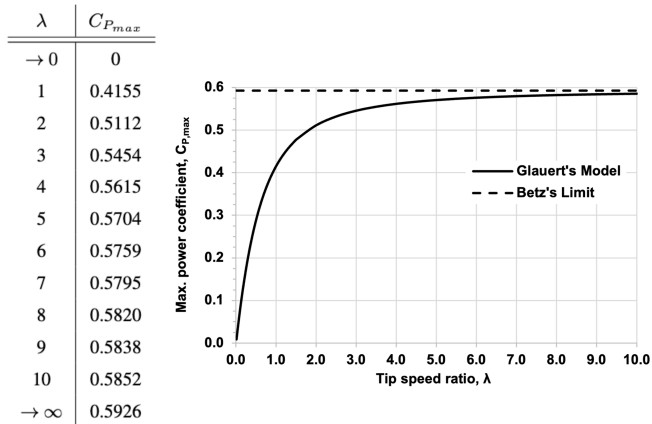

| $\lambda$ | $C_{P_{max}}$ |
|---|---|
| $\to 0$ | 0 |
| 1 | 0.4155 |
| 2 | 0.5112 |
| 3 | 0.5454 |
| 4 | 0.5615 |
| 5 | 0.5704 |
| 6 | 0.5759 |
| 7 | 0.5795 |
| 8 | 0.5820 |
| 9 | 0.5838 |
| 10 | 0.5852 |
| $\to \infty$ | 0.5926 |

**Figure A2.** Maximum power coefficient, $C_P$, for Glauert's actuator disk model and Betz's theoretical limit.

The dotted horizontal line represents the theoretical Betz limit at $\frac{16}{27} = 0.5926$. A more compact form of Glauert's solution can be found in Durand's review (Glauert, 1935) and other texts.

**Data availability.** The data can be provided on request by contacting the authors.

**Author contributions.** DT performed all derivations and wrote the paper. SS advised on derivations and edited the paper.

**Competing interests.** The contact author has declared that neither of the authors has any competing interests.

Note that the limits of integration have been modified to account for the variable substitution from $\lambda_r$ to $a$. The value of the lower bound, $a_1$, can be calculated by setting $\lambda_r$ equal to 0 in Eq. (A8) and solving for $a$ such that $a_1 = \frac{1}{4}$; the upper bound, $a_2$, is the solution to Eq. (A8) for a variable input of $\lambda_r$. Through integration by substitution, a new variable $x = 1 - 3a$ is introduced to allow solving for $C_{P_{max}}$. The exact integral can now be expressed in terms of only $x$.

$$C_{P_{max}} = -\frac{1}{\lambda^2} \cdot \frac{8}{729} \int_{x_1}^{x_2} \left[ \frac{(x+2)(4x-1)(2x+1)}{x} \right]^2 dx \quad (A12)$$

Here the integration bounds must be adjusted to account for the substitution from $a$ into terms of $x$ such that $x_2 = 1 - 3a_2$ and $x_1 = 1 - 3a$. Expanding the integrand and switching the integration bounds to avoid representing the exact integral as a negative expression yields the following:

$$C_{P_{max}} = \frac{1}{\lambda^2} \cdot \left(\frac{2}{9}\right)^3 \int_{x_2}^{x_1} \left[ 64x^4 + 288x^3 + 372x^2 \right.$$

$$\left. + 76x - 63 - \frac{12}{x} + \frac{4}{x^2} \right] dx. \quad (A13)$$

Integration of Eq. (A13) results in

$$C_{P_{max}} = \frac{1}{\lambda^2} \cdot \left(\frac{2}{9}\right)^3 \left[ \frac{64}{5}x^5 + 72x^4 + 124x^3 + 38x^2 - 63x \right.$$

$$\left. - 12\ln x - \frac{4}{x} \right]_{x_2 = 1-3a_2}^{x_1 = \frac{1}{4}}. \quad (A14)$$

These results for $C_{P_{max}}$, representing Glauert's optimum model, are plotted over a range of tip speed ratio, $\lambda$, as a solid line in Fig. A2.

**Disclaimer.** Publisher's note: Copernicus Publications remains neutral with regard to jurisdictional claims made in the text, published maps, institutional affiliations, or any other geographical representation in this paper. While Copernicus Publications makes every effort to include appropriate place names, the final responsibility lies with the authors.

**Acknowledgements.** The authors want to thank the Department of Aerospace Engineering at The Pennsylvania State University for awarding part of this work the Wolk Thesis Award for best research conducted by a senior undergraduate student.

**Review statement.** This paper was edited by Jens Nørkær Sørensen and reviewed by two anonymous referees.

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
