# Peer review of "Glauert's Optimum Rotor Disk Revisited — A Calculus of Variations Solution and Exact Integrals for Thrust and Bending Moment Coefficients"

_Wind Energy Science, 2024_

## Referee Comment (RC1)

The calculus of variations (CoV) is a powerful mathematical tool for the analytic or semi-analytic determination of optimal solutions for a wide range of problems. Its first application to rotors that I am aware of is by Breslin & Andersen (1994, Additional References below) for lightly-loaded propellers. Using the Kutta-Joukowsky form of the thrust equation for an actuator disk representing the rotor, they maximised thrust for a given power and showed the critical importance of the pitch of the trailing vorticity; it is the ratio of torque to thrust. It follows that an optimal rotor has constant pitch throughout its wake. Wood & Hammam (2022) used the converse CoV analysis for wind turbines in which power is maximised for a given thrust. They also used the Kutta-Joukowsky thrust equation which avoids the need to consider pressure in the wake. They removed the restriction to light-loading and showed the same importance of the vortex pitch as the ratio of torque to thrust. No assumptions were made about the uniformity of the velocity through the rotor as is made in the present analysis based on the work of Glauert. The three other important results from Wood & Hammam (2022) were:

1. Their figure 4 shows that the axial induction factor is quadratic in radius at low tip speed ratio, $\lambda$, but becomes approximately constant at higher $\lambda$,
2. The disk loading – the angular velocity behind the disk, which is proportional to the bound circulation - behaves in the same way. As $\lambda \rightarrow 0$, the loading is quadratic in radius whereas it is constant with radius at high $\lambda$,
3. Optimal performance at low $\lambda$ is constrained by the need to avoid recirculation in the wake.

The first two results certainly, and the third possibly, cannot be obtained from the Glauert assumption of constant induction so the present results are valid only for $\lambda$ that is sufficiently high for the axial induction factor to be approximately constant. To quantify the differences in the analyses: Wood & Hammam (2022) found the thrust coefficient $C_T \rightarrow 0.357$ as $\lambda \rightarrow 0$ which is less than half the value in figure 1 of the present submission. Interestingly, the differences in power coefficient are small at $\lambda = 1$: Wood & Hammam (2022) obtained 0.4381 compared to 0.4155 here whereas their $C_T = 0.738$ compares to 0.846 here. Thus, allowing for radial variation in the parameters leads to a more optimal solution at low $\lambda$. To my knowledge, there are, unfortunately, no experimental studies that would help decide the question of optimal performance at low $\lambda$.

Additional References

Breslin, J. P., and Andersen, P. (1994). Hydrodynamics of ship propellers. Cambridge University Press.

$C_T$. Optimal performance of actuator disc models for horizontal-axis turbines. Frontiers in Energy Research, 10, 971177.

---

## Referee Comment (RC2)

**Review of paper:  wes-2024-111:**

**Glauert's Optimum Rotor Disk Revisited – A Calculus of Variations Solution and Exact Integrals for Thrust and Bending Moment Coefficients**

by  Authors: Divya Tyagi and Sven Schmitz

**Brief summary**

The authors present an amendment to Glauert's optimum rotor disk solution for the maximum power coefficient, CPmax, as a function of tip speed ratio, using  a simple alternative solution based on calculus of variations. Next deriving corresponding exact integrals for the thrust coefficient CT and bending moment CBe. Then using L'Hôpital's  rule up to the third derivative of the equations, the thrust coefficient and power coefficient for the tip speed ratio going to zero are determined.

**Overall comments**

Generally, the paper is well written with a logical development of the equations. It seems that the use of the method of calculus of variations yields an elegant approach for derivation of CPmax, CT and CBe for the tip speed ratio going towards infinity and zero, respectively.

However, the paper lacks completely an introduction section giving the work a wider perspective and describing the importance of the work.  In particular, the advancement by previous work mentioned in the paper, "Wilson et al., 1974; Hansen, 2008; Manwell et al., 2009; Burton et al., 2011; Wood, 2011; Schaffarczyk, 2014; Sørensen, 2016; Schmitz, 2019) should be elaborated and  the relation to  the present work should be clear.

**Specific comments and proposal for improvement**

- As mentioned above an introduction section should be inserted with a relevant literature review in relation to the subject. What advancements have been achieved in the past and how the present work contributes to new insight ?
- Discussing shortcomings in the theory where they appear
  - E.g. the jump from equation 29 with the thrust coefficient CT for the whole disc to eq. 30 with the thrust coefficient on an incremental form
    - Is eq. 30 valid in any case ?
- And the shortcomings  and approximations behind the Glauert theory e.g. neglecting the pressure variation in the wake due to swirl

This was discussed at an early stage in the development of the aerodynamic theory for wind turbines by de Vries[1] followed by e.g. Sharp[2] and more recently with a numerical design study[3] that demonstrated a high Cp can be obtained even at low tip speed ratios towards the rotor centre.

As the present study in particular focuses on the power, thrust and bending moment for the tip speed ratio going to zero its proposed to include a discussion of these aspects.

**Final conclusion of review**

The reviewer can recommend publication of the paper but recommends to integrate response to the above review comments.
* * *
[1] Vries, O. de., North Atlantic Treaty Organization, AGARD, NATO, and OTAN. 1979. Fluid Dynamic Aspects of Wind Energy Conversion :

[2] Sharpe, D. J. 2004. "A General Momentum Theory Applied to an Energy-Extracting Actuator Disc." Wind Energy 7 (3): 177–88. https://doi.org/10.1002/we.118.

[3] Johansen, Jeppe, et al. "Design of a Wind Turbine Rotor for Maximum Aerodynamic Efficiency." Wind Energy, vol. 12, no. 3, 2009, pp. 261–73, https://doi.org/10.1002/we.292.

---

## Author Comment (AC1)

**[wes-2024-111]**

Dear Anonymous Referee #1,

Thank you for your detailed review of the paper and provided comments. The authors have acknowledged the necessity of including a small section discussing the validity of Glauert's rotor disk theory, particularly at low tip speed ratios. Below are the author responses in *italics* and Anonymous Referee #1 comments in blue.

The calculus of variations (CoV) is a powerful mathematical tool for the analytic or semi-analytic determination of optimal solutions for a wide range of problems. Its first application to rotors that I am aware of is by Breslin & Andersen (1994, Additional References below) for lightly-loaded propellers. Using the Kutta-Joukowsky form of the thrust equation for an actuator disk representing the rotor, they maximised thrust for a given power and showed the critical importance of the pitch of the trailing vorticity; it is the ratio of torque to thrust. It follows that an optimal rotor has constant pitch throughout its wake. Wood & Hammam (2022) used the converse CoV analysis for wind turbines in which power is maximised for a given thrust. They also used the Kutta-Joukowsky thrust equation which avoids the need to consider pressure in the wake. They removed the restriction to light-loading and showed the same importance of the vortex pitch as the ratio of torque to thrust. No assumptions were made about the uniformity of the velocity through the rotor as is made in the present analysis based on the work of Glauert. The three other important results from Wood & Hammam (2022) were:

> 1. Their figure 4 shows that the axial induction factor is quadratic in radius at low tip speed ratio, $\lambda$, but becomes approximately constant at higher $\lambda$,

> 2. The disk loading – the angular velocity behind the disk, which is proportional to the bound circulation - behaves in the same way. As $\lambda \to 0$, the loading is quadratic in radius whereas it is constant with radius at high $\lambda$,

> 3. Optimal performance at low $\lambda$ is constrained by the need to avoid recirculation in the wake.

The first two results certainly, and the third possibly, cannot be obtained from the Glauert assumption of constant induction so the present results are valid only for $\lambda$ that is sufficiently high for the axial induction factor to be approximately constant.

*Author Response: The authors have added Section 6.1 titled Validity of Classical Glauert Theory in the revised manuscript. Glauert's assumptions in solving the 1-D problem are listed both in the introduction and section 6.1. The authors also acknowledge the approach taken by Wood & Hammam.*

To quantify the differences in the analyses: Wood & Hammam (2022) found the thrust coefficient CT → 0.357 as $\lambda \to 0$ which is less than half the value in figure 1 of the present submission. Interestingly, the differences in power coefficient are small at $\lambda = 1$: Wood & Hammam (2022) obtained 0.4381 compared to 0.4155 here whereas their CT = 0.738 compares to 0.846 here. Thus, allowing for radial variation in the parameters leads to a more optimal solution at low $\lambda$. To my knowledge, there are, unfortunately, no experimental studies that would help decide the question of optimal performance at low $\lambda$.

*Author Response:* The authors have added more details in Section 6.1 titled *Validity of Classical Glauert Theory* of the revised manuscript. It is evident that Wood & Hammam's approach yields more correct solutions at low $\lambda$, however it is restated that the purpose of this work was to simply build upon Glauert's classical theory, including all the associated assumptions.

Additional References

Breslin, J. P., and Andersen, P. (1994). Hydrodynamics of ship propellers. Cambridge University Press.

CT. Optimal performance of actuator disc models for horizontal-axis turbines. Frontiers in Energy Research, 10, 971177.

*Author Response:* The authors have added the suggested and other references (10 total) to the manuscript.

---

## Author Comment (AC2)

**[wes-2024-111]**

Dear Anonymous Referee #2,

Thank you for your detailed review of the paper and the provided comments for improvement. The authors acknowledge the need for including a brief relevant literature review discussing rotor disk theory. Below are the author responses in *italics* and Anonymous Referee #2 comments in blue.

Generally, the paper is well written with a logical development of the equations. It seems that the use of the method of calculus of variations yields an elegant approach for derivation of $C_{Pmax}$, $C_T$ and $C_{Be}$ for the tip speed ratio going towards infinity and zero, respectively.

However, the paper lacks completely an introduction section giving the work a wider perspective and describing the importance of the work. In particular, the advancement by previous work mentioned in the paper, "Wilson et al., 1974; Hansen, 2008; Manwell et al., 2009; Burton et al., 2011; Wood, 2011; Schaffarczyk, 2014; Sørensen, 2016; Schmitz, 2019) should be elaborated and the relation to the present work should be clear.

As mentioned above an introduction section should be inserted with a relevant literature review in relation to the subject. What advancements have been achieved in the past and how the present work contributes to new insight?

*Author Response: Thank you for the positive comments about the writing and mathematics derived in the paper. The authors agree that an introduction section is helpful to highlight developments in optimum rotor disk models over the years. This also provides background for the new insights and coefficients derived in the present work within the constraints of Glauert's theory. Please see Section 1 titled Introduction in the revised manuscript. A total of 10 additional references were added.*

Discussing shortcomings in the theory where they appear

- E.g. the jump from equation 29 with the thrust coefficient CT for the whole disc to eq. 30 with the thrust coefficient on an incremental form

  - Is eq. 30 valid in any case?

*Author Response: Very good comment, thank you. Additional steps have been depicted showing the derivation of the incremental thrust coefficient from the original definition of $C_T$. This math can be found in the beginning of Section 4 titled Exact Integral of the Thrust Coefficient $C_T$ Based on Glauert's Optimum Solution, specifically in Eqs. 29-31. Eq. 31 (previously Eq. 30) is valid for defining the increment in thrust coefficient between blade sections for all $\lambda > 0$.*

And the shortcomings and approximations behind the Glauert theory e.g. neglecting the pressure variation in the wake due to swirl

*Autor Response: The authors agree that some discussion is useful to the present work; Section 6.1 titled Validity of Classical Glauert Theory has been added to the revised manuscript. Also, Glauert's inherent assumptions are now mentioned in the introduction in the context of other works.*

This was discussed at an early stage in the development of the aerodynamic theory for wind turbines by de Vries[1] followed by e.g. Sharp[2] and more recently with a numerical design study that demonstrated a high $C_p$ can be obtained even at low tip speed ratios towards the rotor centre.

As the present study in particular focuses on the power, thrust and bending moment for the tip speed ratio going to zero its proposed to include a discussion of these aspects.

*Author Response:* The authors have addressed other works related to rotor disk theory in the introduction section, and have added some relevant references. In particular, the results at low tip speed ratio have been brought into context with recent work by Wood & Hammam (2022). Note that the objective of this work is simply an addendum to Glauert and not a proposal for improved performance coefficients at very low tip speed ratio. It is noteworthy, however, that some quantitative comparisons to Wood & Hammam (2022) show that results obtained using Glauert's theory are quite good at low tip speed ratio given the inherent assumptions, see Section 6.1.

**Final conclusion of review**

The reviewer can recommend publication of the paper but recommends to integrate response to the above review comments.

*Author Response:* Thanks again for the very good comments. The authors feel that we addressed the helpful suggestions, also in conjunction with comments made by the other reviewer. In the end, this work is an interesting closed-form analytical addition to a fundamental work in wind turbine aerodynamics.

---

## Author Response (AR2)

Dear Associate Editor,

 Dear Reviewers,

We would like to thank you for a thorough and timely review process. Your comments made the paper better and provide more context for Glauert's work within generalized momentum theories.

With best regards,

 Divya Tyagi & Sven Schmitz